# Kondo quasiparticle dynamics observed by resonant inelastic x-ray scattering

M. C. Rahn [1,2] ✉, K. Kummer [3], A. Hariki [4,5], K.-H. Ahn [5,6], J. Kuneš [5], A. Amorese [7,8], J. D. Denlinger [9], D.-H. Lu [10], M. Hashimoto [10], E. Rienks [11], M. Valvidares [12], F. Haslbeck [13,14], D. D. Byler [1], K. J. McClellan [1], E. D. Bauer [1], J. X. Zhu [1], C. H. Booth [15], A. D. Christianson [16], J. M. Lawrence [1,17], F. Ronning [1] & M. Janoschek [1,14,15,18,19] ✉

Effective models focused on pertinent low-energy degrees of freedom have substantially contributed to our qualitative understanding of quantum materials. An iconic example, the Kondo model, was key to demonstrating that the rich phase diagrams of correlated metals originate from the interplay of localized and itinerant electrons. Modern electronic structure calculations suggest that to achieve quantitative material-specific models, accurate consideration of the crystal field and spin-orbit interactions is imperative. This poses the question of how local high-energy degrees of freedom become incorporated into a collective electronic state. Here, we use resonant inelastic x-ray scattering (RIXS) on $CePd_3$ to clarify the fate of all relevant energy scales. We find that even spin-orbit excited states acquire pronounced momentum-dependence at low temperature—the telltale sign of hybridization with the underlying metallic state. Our results demonstrate how localized electronic degrees of freedom endow correlated metals with new properties, which is critical for a microscopic understanding of superconducting, electronic nematic, and topological states.

Metals containing lanthanides are optimally suited to study the Kondo lattice problem, owing to highly localized $4f$-electrons, which at low temperature form a strongly-correlated many-body state[1,2]. The Kondo lattice model is characterized by two effective low-energy scales[3] (see Fig. 1a–c): the Kondo temperature $T_K$, below which local $f$-electron magnetic moments become effectively coupled to the surrounding conduction electrons, and the coherence temperature $T_{coh}$, below which these states are coherently incorporated into the underlying band structure as dispersive, albeit heavy, electronic quasiparticles. A large body of experiments has shown that real materials can indeed be qualitatively characterized by these two energy scales. Modern angle-resolved photoemission spectroscopy

[1]Los Alamos National Laboratory, Los Alamos, NM 87545, USA. [2]Institute for Solid State and Materials Physics, Technical University of Dresden, 01062 Dresden, Germany. [3]European Synchrotron Radiation Facility, BP 220, F-38043 Grenoble Cedex, France. [4]Department of Physics and Electronics, Graduate School of Engineering, Osaka Prefecture University, Sakai, Osaka 599-8531, Japan. [5]Institute for Solid State Physics, TU Wien, 1040 Vienna, Austria. [6]Institute of Physics of the CAS, Cukrovarnická 10, 162 00 Praha 6, Czechia. [7]Institute of Physics II, University of Cologne, Cologne, Germany. [8]Max Planck Institute for Chemical Physics of Solids, Dresden, Germany. [9]Advanced Light Source, Lawrence Berkeley Laboratory, Berkeley, CA 94720, USA. [10]Stanford Synchrotron Radiation Lightsource, SLAC National Accelerator Laboratory, Menlo Park, CA 94025, USA. [11]Helmholtz Zentrum Berlin, Bessy II, D-12489 Berlin, Germany. [12]ALBA Synchrotron Light Source, E-08290 Cerdanyola del Vallès, Barcelona, Spain. [13]Physik-Department, Technische Universität München, D-85748 Garching, Germany. [14]Institute for Advanced Studies, Technische Universität München, D-85748 Garching, Germany. [15]Chemical Sciences Division, Lawrence Berkeley National Laboratory, Berkeley, CA 94720, USA. [16]Oak Ridge National Laboratory, Oak Ridge, TN 37831, USA. [17]Department of Physics and Astronomy, University of California, Irvine, CA 92697, USA. [18]Laboratory for Neutron and Muon Instrumentation, Paul Scherrer Institute, CH-5232 Villigen, Switzerland. [19]Physik-Institut, Universität Zürich, Winterthurerstrasse 190, CH-8057 Zürich, Switzerland. ✉e-mail: marein.rahn@tu-dresden.de; marc.janoschek@psi.ch

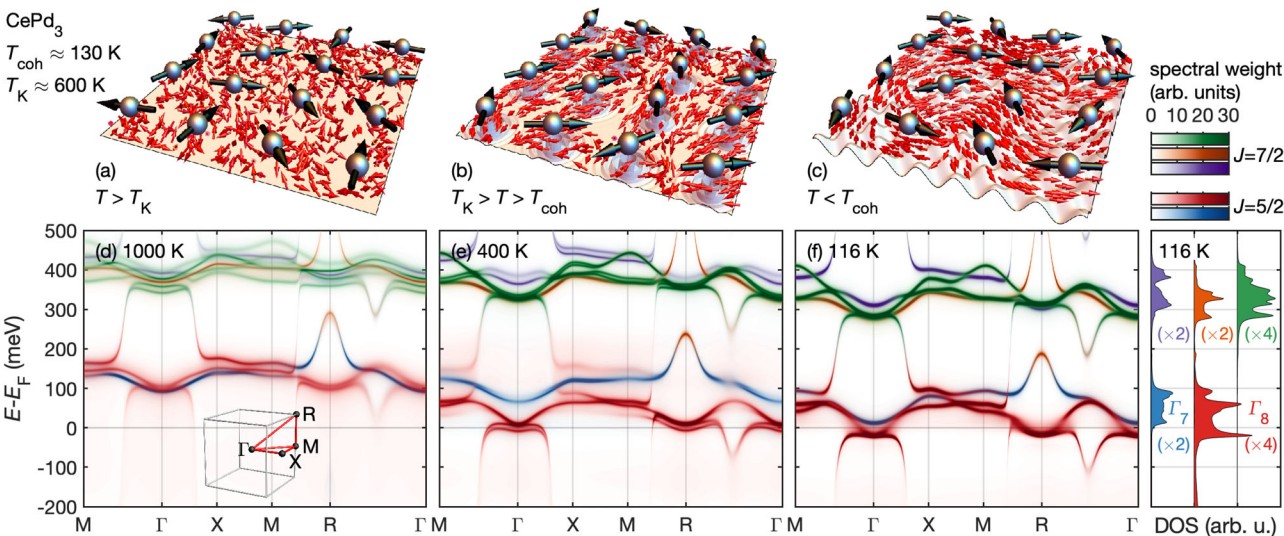

**Fig. 1 | Emergence of a coherent Kondo lattice state as a function of temperature in CePd₃. a–c** Real-space impressions of the increasingly coherent screening of Ce moments (black arrows) across the temperature scales $T_K$ and $T_{coh}$. Spins of itinerant electrons are represented by red arrows (see text for details). **d–f** Electronic spectral function $A(k, \omega)$ of CePd₃ obtained from the combination of density functional and dynamical mean field theory (DFT+DMFT) at 1000 K, 400 K, and 116 K. The fourteen $4f$ orbitals of cerium form seven Kramers doublets, separated into a $J = 5/2$ sextuplet and a $J = 7/2$ octuplet by $E_{SO} \approx 280$ meV. A cubic crystal field (CF) further induces a $E_{SO} \approx 10$ meV splitting between a $J = 5/2\,\Gamma_7$ doublet (blue) and the $\Gamma_8$ quartet (red). The $J = 7/2$ states are split into two doublets (purple and orange) and one quartet (green). **a, d** Above the Kondo temperature, $T_K \approx 600$ K, local $f$-electron moments are uncorrelated. The corresponding electronic structure only shows weak hybridization with the incoherent $f$-states. **b, e** For $T < T_K$, local $f$-electron magnetic moments effectively couple to the surrounding conduction electrons, forming virtual-bound states in the vicinity of the Fermi energy $E_F$ (suggested as ripples in **b**). The $f$ spectral weight close to the Fermi surface remains predominantly incoherent. **c, f** Below the coherence temperature $T_{coh} \approx 130$ K, the $f$-states are coherently incorporated (suggested by the lattice-coherent modulation in Panel **c**) into the underlying band structure as dispersive, albeit heavy, electronic quasiparticles. The right panel shows the density of states (DOS) summed over the displayed high-symmetry directions, projected onto the different CF characters.

(ARPES)[4], quasiparticle interference[5], and inelastic neutron scattering (INS)[6] have even directly observed the formation of coherent heavy quasiparticle bands. Frustrated magnetism[7], unconventional superconductivity[8], hidden order[9], as well as topologically non-trivial[10–12] and electronic-nematic states[13,14] are all known to emerge from this Kondo lattice state.

The two phenomenological energy scales ($T_K$, $T_{coh}$) clearly do not suffice to account for this variety of experimental discoveries. Instead, material-specific knowledge of the quasiparticle fine structure will be required to disentangle the underlying interaction channels and achieve a guided design of emergent functional properties. Magnetic, thermodynamic[15], and X-ray spectroscopic[16] studies of crystal-field (CF) ground states in Kondo lattices have indeed demonstrated that the anisotropic hybridization of the $f$-electron wave function strongly influences the ground state properties, even for materials with the same crystal structure. Moreover, bulk measurements as a function of external tuning parameters like magnetotransport and magnetostriction suggest that the structure of unoccupied crystal field levels cannot be disregarded in these investigations[17,18]. Realistic models of these circumstances can be achieved by the combination of density functional and dynamical mean field theory (DFT+DMFT). And indeed, such calculations have recently confirmed that high energy scales like CF and SO splitting must be taken into account to quantitatively reproduce low-energy ground states[4,19,20].

Clarifying the mechanisms by which the properties of localized $f$-electrons blend into an itinerant environment is an outstanding experimental challenge. While local probes capture the overall energy scales of CF and SO excited states, they cannot observe their momentum dependence—the telltale sign of hybridization with the underlying metallic state. This crucial microscopic knowledge requires low-energy momentum-resolved spectroscopic methods like quasiparticle interference or INS. Recent pioneering ARPES studies[4,21,22] have directly mapped anisotropic hybridization close to

the Fermi energy in CeMIn₅ and CeM₂X₂ heavy fermion materials ($M$ = transition metal; $X$ = Si, Ge). This new level of material-specific microscopic insights has shone a light on the relevance of crystal field and magnetic degrees of freedom, and has called into question widely accepted concepts like the direct relation between the temperature scale $T_{coh}$ and the formation a "large" Fermi surface[22]. One common constraint of these techniques is that they lack sensitivity to the unoccupied $f$-electronic structure at higher energies[23]. To overcome this limitation, we take advantage of instrumental advances that have made resonant inelastic X-ray scattering (RIXS) a powerful probe of Kondo lattices[24,25].

$M$-edge RIXS is a photon-in/photon-out process in which $3d$ core electrons are resonantly excited and recombine by dipole-transitions after interacting with $4f$ valence states. Like INS, this serves as a probe of two-particle correlations, providing a complementary perspective on the single-particle spectral function measured by ARPES. It has the advantage of true bulk-sensitivity at good energy resolution, but, in contrast to INS, the observation of inter-/intraband transitions by X-rays is not restricted to the spin-flip channel. RIXS also has major experimental advantages, as it is not dominated by phonon scattering, and does not require large single crystals and long counting times. Crucially for the present context, the insights are neither limited by the instrumental energy resolution, nor in the range of energy transfer, and the resonant character and polarization dependence allow to separate different excitation channels[25].

We focus on the archetypal Kondo lattice material CePd₃, with an effective $f$-occupation of $n_f \approx 0.75$ (from X-ray absorption spectroscopy[26]), and $k_B T_K \approx 55$ meV. The Kondo energy of this system is thus intermediate to the bare SO coupling $E_{SO} \approx 280$ meV of Ce and the coherence energy scale $k_B T_{coh} \approx 11$ meV (derived from transport data)[27] and CF splitting $E_{CF} \approx 10$ meV (from comparison to the isostructural material CeIn₃[28] and the present DFT calculation). This cascade of energy scales makes CePd₃ ideally suited to address

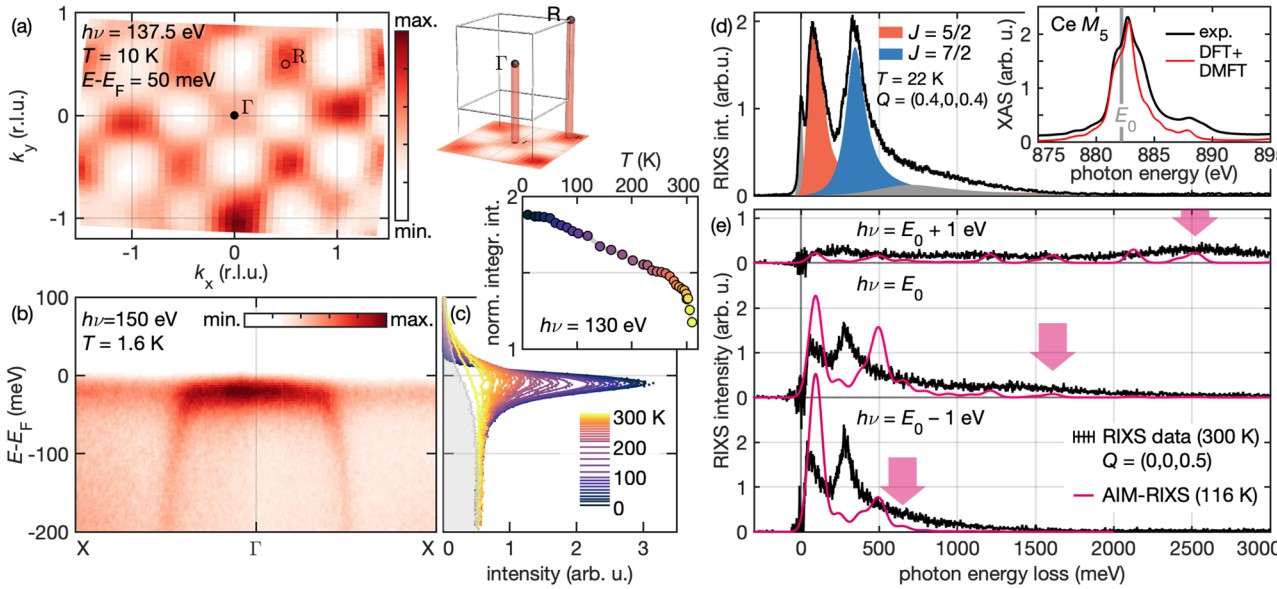

**Fig. 2 | Comparison of our model with spectroscopic data, covering the electronic structure of CePd$_3$ from the meV to the eV scale.** For a global comparison of the DFT+DMFT calculation with the prototypical Kondo lattice material, we employ three spectroscopic methods. **a**–**c** Vacuum ultraviolet angle-resolved photo emission spectroscopy (ARPES) reveals the electronic spectral function $A(k, \omega)$ with meV resolution. **a** Low-temperature quasiparticle Fermi surface observed by ARPES. As the momentum along the surface normal is not resolved, the pockets Γ and R are projected onto the $k_x$–$k_y$ plane (cf. illustration of the Brillouin zone). **b** High-resolution ARPES spectrum highlighting the hybridization of the almost vertical Pd 4$d$ bands with the Ce 4$f$ states at the Γ point. **c** Temperature dependence of energy distribution curves of this Kondo resonance. The gray shaded area is the reference spectrum of a gold foil at 310 K. The inset shows the integrated intensity of $f$-electronic density of states normalized to the integrated intensity of the metallic background, as approximated by the gold spectrum.

**d**, **e** Resonant inelastic X-ray scattering (RIXS) covering energy transfers of several eV. **d** Overview of the characteristic features of CePd$_3$ RIXS at $E_0 = 882.2$ eV, here shown for $Q = (0.4, 0, 0.4)$, at 22 K. Spectral weight assigned to excitations of either spin orbit state is marked by colored Lorentzian lineshapes. The resolution-limited elastic peak and a sloping background due to excitations into the continuum of Pd 4$d$ states are shaded gray. The inset shows a comparison of the low-temperature Ce $M_5$ X-ray absorption spectroscopy (XAS) spectrum with our DFT+DMFT calculation. **e** Comparison of a $Q = (0, 0, 0.5)$ RIXS spectrum at $E_0$ with data recorded 1 eV above/below this resonance (elastic line subtracted). Pink lines show the respective results of the AIM-RIXS calculation at 116 K. Aside from the Raman-like $J = 5/2$ and $J = 7/2$ peaks, the calculation captures the shift of continuum excitations to higher energy transfers at higher photon energies $h\nu$ (indicated by arrows). Due to the necessarily coarse discretization, the SO gap is consistently overestimated and the continuum is mapped onto a series of peaks.

both our experimental and computational objectives, i.e., to understand how crystal field and spin-orbit interactions become coherently incorporated into the Kondo lattice state. The relatively large $T_{coh}$ and $T_K$ ensure that temperatures accessible by DFT+DMFT are sufficient to simulate coherence[6], and the quasiparticle bandwidth is large compared to the energy resolution d$E$ ~ 35 meV of RIXS.

## Results

Figure 1 shows the correlated band structure of CePd$_3$ at 1000 K, 400 K, and 116 K, as obtained via DFT+DMFT calculations to visualize the emergence of a coherent Kondo lattice in CePd$_3$. Cerium features fourteen 4$f$ orbitals corresponding to seven Kramers doublets, separated into a $J = 5/2$ sextuplet and a $J = 7/2$ octuplet by $E_{SO}$. The cubic CF further splits the $J = 5/2$ and $J = 7/2$ states into a total of five CF multiplets indicated in Fig. 1 in distinct colors. Figure 1d illustrates that even at $T = 1000$ K $> T_K$, the modulation of the 4$f$ states due to the onset of anisotropic hybridization outweighs the weak CF splitting, which underscores the importance of momentum-resolved data. Below $T_K \approx$ 600 K, the increasing hybridization renormalizes all 4$f$-states. However, their spectral weight close to the Fermi surface remains predominantly incoherent (Fig. 1e). For $T < T_{coh} \approx 130$ K, coherence sets in, and well-defined quasiparticle bands that cross the Fermi level form pockets at the Γ and R points (Fig. 1f). Our calculations emphasize the amount of information encoded in the unoccupied states. Instead of the Γ$_7$ CF ground state of isostructural (and more localized) CeIn$_3$[29], we find that the hybridization with Pd $d$ bands endows the quasiparticle pockets with Γ$_8$-like character akin to CeB$_6$[30] (for CePd$_3$ at 116 K we obtain occupation numbers $n_{Γ7} = 0.08$ and $n_{Γ8} = 0.68$). The SO excited

states are similarly modified due to hybridization. Figure 1f shows the summed density of states of different orbital characters. We find that the $J = 7/2$-like states retain an occupation number of 0.17, even at low temperatures (116 K). This indicates that, aside from the mixing of CF states, strictly speaking, even $J$ cannot be considered a good quantum number ($E_K \approx 0.2 E_{SO}$).

Recently, Goremychkin et al.[6] used state-of-the-art DFT+DMFT calculations, like those presented here, to quantitatively model their neutron spectra of magnetic inter-band transitions and conclusively demonstrate the formation of coherent quasiparticle bands in CePd$_3$. This pioneering work has raised several remaining key issues on the nature of the coherent state: (i) To date, the phenomenological temperature scale $T_{coh}$ has only been inferred from bulk properties like maxima in resistivity and magnetic susceptibility[27] and the onset of phase-coherence remains to be explored microscopically. (ii) The INS study[6] was limited to energy transfers between 20 and 65 meV, but also predicted excitations within the putative SO gap, which calls into question how coherence impacts the coupling of spin and orbital momentum. (iii) To clarify the fate of CF ground states assigned to heavier Ce systems, it would be crucial to resolve the fine structure of the quasiparticle bands.

To address these challenges, our study has to overcome the limitations of INS as a probe of electronic structure and encompass all energy scales relevant for DFT+DMFT. To complement RIXS, we present first single-crystal ARPES data of CePd$_3$, resolving electronic structure within few meV of the Fermi level. Moreover, we corroborate the two-particle dynamics in the 50–250 meV regime by higher-energy INS (see Section 3 of the Supplementary Information (SI)[31] file), and use X-ray absorption spectroscopy (XAS) to probe the RIXS intermediate

states in the presence of a Ce $3d_{5/2}$ core hole. This provides fixed points for a global, realistic model of the inter-band transitions.

Figure 2a, b shows the occupied quasiparticle states in the vicinity of the Fermi level of a [1, 0, 0]-fractured sample at low temperatures obtained via single-crystal vacuum ultraviolet ARPES measurements. As predicted by DFT+DMFT, quasiparticle Fermi pockets form at the Γ and R points. Since the electron momentum perpendicular to the sample surface ($k_z$) is not resolved in these measurements, both Γ and R hot spots are projected onto the $k_x$–$k_y$ plane. The observed characteristic checkerboard-pattern of the dispersion around the Γ point confirm that DFT+DMFT yields an adequate model of the coherent Kondo lattice state with regard to the quasiparticles below the Fermi energy. Figure 2c illustrates how spectral weight due to increasing hybridization of $f$-states continuously rises at low temperatures and forms a ~30 meV wide Kondo peak at the Fermi surface, which is closely reminiscent of Fig. 1d–f.

Now we turn to discuss our RIXS measurements. Figure 2d shows a typical $M_5$ RIXS spectrum of CePd$_3$, recorded at 22 K (i.e., well below the coherence temperature), at a photon energy of $E_0 = 882.2$ eV, where excitations of $f$-quasiparticles with $J = 5/2$ and $J = 7/2$-like character carry similar spectral weight (as indicated by phenomenological colored Lorentzian line shapes). The two SO bands are much wider than the instrumental energy resolution, which can be inferred from the elastic line, a Gaussian peak of 32 meV full-width at half-maximum. Their intensity appears in the crossed ($\sigma\pi'$) polarization channel and is likely of magnetic character (see Supplementary Note 6[31]). We find no additional two-particle excitation in the ~200 meV regime. Instead, up to this energy, RIXS is in quantitative agreement with the dynamic magnetic susceptibility $\chi''$ ($Q, \omega$) probed by INS (see our detailed discussion in Supplementary Note 3[31]). However, neutron excitations across the spin-orbit gap are suppressed by a factor ~200 [23], which makes it de facto impossible to probe momentum dependence of excited $f$-electron states with INS. As a combination of two consecutive dipole transitions, RIXS breaks free of these constraints. Moreover, by selecting the intermediate state and using polarization analysis, it is possible to separate the $J = 5/2$ and $J = 7/2$ RIXS spectral weight[31].

For a realistic model of RIXS in CePd$_3$ going beyond the comparison to INS, we compute the Kramers-Heisenberg cross section in an Anderson impurity model based on the DFT+DMFT electronic structure[31]. Notably, this AIM-RIXS calculation relies on a single parameter, the double-counting correction $\mu_{dc}$, which shifts the energy of the $4f$ states to take into account the $f$–$f$ interaction present in the DFT calculation. The value of $\mu_{dc}$ was chosen to reproduce the XAS characteristics, shown in the inset to Fig. 2d, and the density of states observed in direct and inverse photoemission[31]. It is further corroborated by the favorable comparison with ARPES. We thus achieve a reliable model of the electronic structure spanning three orders of magnitude in energy, from the $f$–$f$ interaction ($U_{ff} = 6$ eV) down to the width of the occupied $f$-states at the Fermi surface (~30 meV) and CF splitting (10 meV).

The exact evaluation of the Kramers-Heisenberg term in the AIM comes at the price of losing the momentum-resolution and of discretizing the DFT+DMFT bath (we use a set of 20 levels[31]). Subtle aspects of the electronic structure, like an accurate reproduction of the renormalized spin-orbit gap are lost in this necessarily coarse model of local-itinerant hybridization. On the other hand, AIM-RIXS does capture the resonant creation of the excited states. In Fig. 2e, we compare calculated spectra with experimental data above and below the resonance at $E_0$. Both quasiparticle excitations behave Raman-like, i.e., the energy transfer is independent of $h\nu$. Excitations of $f$-quasiparticles that escape into the wide Pd $4d$ bands contribute a fluorescence tail that shifts to higher energies at higher photon energies. This effect is well captured in Fig. 2d, even if the continuum is mapped onto a set of discrete peaks. Thorough discussions of the photon-energy

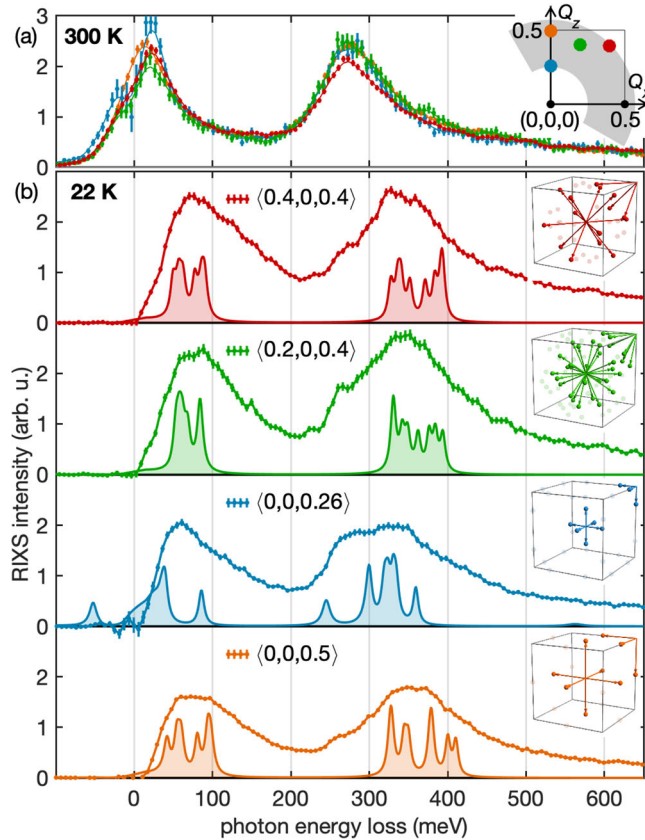

**Fig. 3 | Fingerprint of a coherent Kondo lattice state in CePd$_3$ as observed by resonant inelastic X-ray scattering (RIXS).** RIXS spectra are shown at four different momentum transfers $Q$, at **a** 300 K and **b** 22 K. For better comparison, the quasielastic peak has been subtracted. The positions of $Q$ are indicated in the reciprocal space map inset, where the $Q$ range accessible to $M_5$-edge RIXS is shaded gray. While at $T = 300$ K, well above the coherence temperature $T_{coh} \approx 130$ K, the signal is almost invariant with regard to $Q$, for $T = 22$ K a pronounced momentum-dependence emerges. The inset views of the cubic Brillouin zone in **b** illustrate how the $Q$ positions probed with RIXS correspond to momentum transfers exciting heavy quasiparticles from electron pockets at Γ and R into unoccupied parts of the electronic structure. For reference, the shaded spectra in each panel show the DFT+DMFT density of states, as shown in Fig. 1c, summed over the relevant positions of quasiparticle momentum-transfers in the Brillouin zone.

dependence, polarization dependence, and XAS characteristics are provided in the SI[31].

To probe the momentum-dependence of these $f$-states for $T < T_{coh}$, as predicted in Fig. 1, we recorded RIXS spectra at four different points in the Brillouin zone, at 300 K and at 22 K. As illustrated in Fig. 3a, the response at $T > T_{coh}$ is close to isotropic. By contrast, the corresponding spectra in the lattice-coherent state [Fig. 3b] vary dramatically in both spectral weight and energy. Surprisingly, these observations are most pronounced among the spin-orbit excited quasiparticles, with a dispersion of $\approx 40$ meV between $Q = (0, 0, 0.26)$ and $(0, 0, 0.5)$. In the absence of momentum-resolved RIXS calculations, it is instructive to relate these observations to the DFT+DMFT spectra. Because the occupied $f$-quasiparticles are constrained to narrow hot-spots at Γ and R, a simple argument can be made: For each of the investigated momentum transfers, symmetry-equivalent excitation paths within the Brillouin zone are illustrated in the insets to Fig. 3b. For instance, $Q = (0, 0, 0.5)$ probes transitions (Γ → X, R → M), which should be comparable to $Q = (0.5, 0, 0.5)$ (Γ → M, R → X). This is consistent with the observation of similar spectra at $Q = (0, 0, 0.5)$ and $Q = (0.4, 0, 0.4)$, and significant variations between $Q = (0, 0, 0.26)$ and $(0, 0, 0.5)$. For direct comparison, the sum of the 116 K DFT+DMFT density of states at the

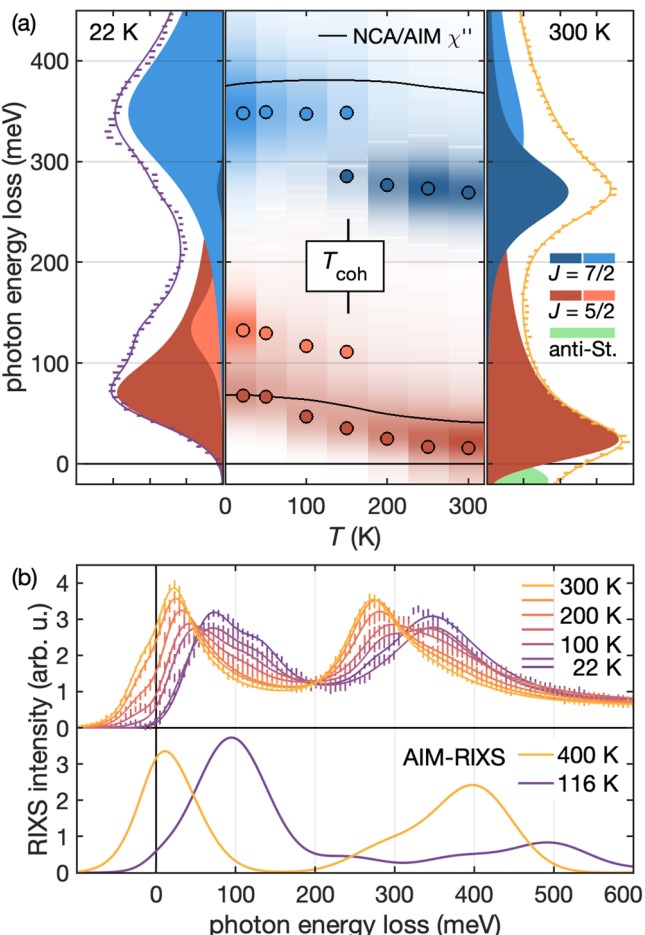

**Fig. 4 | Onset of coherence in CePd₃ as observed by resonant inelastic X-ray scattering (RIXS). a** Detailed thermal variation of RIXS spectra between 22 K and 300 K. For clarity, the elastic lines have been subtracted from the data. The emergence of the coherent state at $T_{coh}$ appears as a transfer of intensity between distinct excitations. We phenomenologically model this by Lorentzian fits of the data, as indicated in the left (22 K) and right (300 K) panels. Colored markers in the center panel indicate the centroids of these features. As the $J = 5/2$ quasiparticles become thermally populated, an anti-Stokes peak (green) appears on the photon-energy-gain side. The black lines show the thermal evolution of the $J = 5/2$ and $J = 7/2$ peaks in our NCA/AIM calculation of $\chi''$ (see text for details). **b** Comparison of temperature-dependent RIXS (top) with the exact AIM-RIXS calculation (bottom). This model accurately captures the parallel shift of $J = 5/2$ and $J = 7/2$ features. The overestimation of the SO gap is an artifact of the necessary discretization (see text for details).

respectively relevant positions in the Brillouin zone is drawn as shaded line shapes in Fig. 3b. Despite $E_K \gg E_{CF}$, RIXS is clearly able to resolve the fine structure of the hybridized spin-orbit multiplets.

As shown in Fig. 4, the characteristics of the RIXS Kondo excitations change dramatically when thermal fluctuations destroy the lattice-coherence. Figure 4a shows spectra at momentum transfer $Q = (0.4, 0, 0.4)$, at 22 K and at 300 K [left and right panels]. Colored lineshapes illustrate phenomenological fits of Lorentzian lineshapes (for better comparison with AIM-RIXS spectra, the elastic line has been subtracted). The center panel shows how, at low temperatures, both excitations continuously shift to higher energies. The black lines indicate the maxima of the momentum averaged dynamic magnetic susceptibility $\chi''(\omega)$ of the Anderson impurity model in the non-crossing approximation (NCA/AIM, see Supplementary Note 1[31]). It is thus evident that even phenomenological models reproduce the basic thermal trend of the $J = 5/2$ excitations. However, at intermediate temperatures, both $J = 5/2$ and $7/2$ bands develop the

momentum-dependent substructure discussed above, which corresponds to the onset of coherence. This microscopic observation of $T_{coh} \approx 130$ K is consistent with bulk methods[27]. Figure 4b shows that our AIM-RIXS approach does indeed capture the parallel thermal shift of both SO states by $\approx 80$ meV. The apparent overestimation of the SO gap is due to the limited number of bath levels determined by computational cost, which does not allow to accurately capture the renormalization of $J = 7/2$ and $5/2$ due to hybridization with conduction bands.

## Discussion

Our study showcases that RIXS provides a comprehensive picture of the emergent coherent Kondo lattice state, enabling a detailed comparison with material-specific state-of-the-art band structure calculations on all relevant energy scales. Crucially, RIXS reveals the exact position of all $f$-electron levels and how they blend into the underlying metallic state. We have used this capability to clarify the characteristic energy scales of the model Kondo lattice material CePd₃. The fine structure and momentum dependence of the observed spectra is consistent with the five CF multiplets inferred from our DFT+DMFT calculation [Fig. 1a]. Below the coherence temperature $T_{coh} \approx 130$ K, which we have determined microscopically, the hybridization with the conduction electrons impacts all $f$ states on the scale of $E_K \approx 55$ meV. Even though the overall 80 meV renormalization of the $J = 5/2$ peak as a function of temperature in Fig. 4a amounts to 30% of the free-ion spin-orbit coupling (280 meV), the SO gap remains unchanged in the coherent Kondo lattice state. This implies that the strength of the SO interaction determined by $\mathbf{L} \cdot \mathbf{S}$ is not impacted by the substantial renormalization of the electronic state that is evidenced by overall shifts of the $f$-levels (cf. Fig. 4) and their variation throughout the Brillouin zone (cf. Fig. 1), both of which are on the order of $E_K$. Interestingly, this observation is contrary to the interpretation by Murani et al.[32], who expected that $f$ band formation should lead to at least a partial quenching of orbital momentum and a reduction of $\mathbf{L} \cdot \mathbf{S}$.

In conclusion, our study reveals in unprecedented detail how the full manifold of localized $f$-electron states becomes hybridized with the underlying band structure in the archetypal Kondo metal CePd₃. Such insight is crucial for the development of fully microscopic and material-specific models of quantum-coherency and to understand how it drives the emergence of novel order parameters. The remarkable discovery that $f$-electron band formation below the coherence temperature does not quench the spin-orbit interaction showcases how—via the mechanism of hybridization—even excited $f$-electron states can endow a material with novel properties that cannot be derived simply from the symmetry of the underlying crystal lattice. This is key to a growing number of topical phenomena such as $p$-wave superconductivity, electronic-nematic order, and topologically non-trivial phases. Given that the present insights were not limited by the experimental energy resolution, similar studies are already feasible in materials with lower Kondo energy scales or in systems where Kondo dynamics can be tuned by external parameters. They can also be performed on microscopic single crystals not accessible to INS and will benefit from continuous improvements in brilliance and energy resolution[33].

## Methods
### Samples
An ingot of cubic CePd₃ (AuCu₃ structure, space group $Pm\bar{3}m$, lattice parameter $a = 4.13$ Å) was grown from a self-contained melt using a modified Czochralski method in a tri-arc furnace. Single-crystalline grains were identified and oriented by backscattering Laue X-ray diffraction. Cuboid samples with faces parallel to [100] planes were then cut from this grain. Incisions were made around the circumference of these samples to favor fracturing parallel to these planes.

## Photoemission spectroscopy

Vacuum ultraviolet angle-resolved photoemission spectroscopy was performed at a BL5-2, SSRL (Stanford), beamline 4.0.3, ALS (Berkeley), and one-cube, BESSY-II (Berlin). The samples were fractured under ultra-high vacuum conditions of $<5 \times 10^{-11}$ Torr. The best contrast from the Ce $4f$ states was obtained above the N ($4d \rightarrow 4f$) absorption edge, at photon energies of 130–150 eV. The electrons were analyzed using Scienta R8000 (4.0.3) and Scienta DA30L (BL5-2, one-cube) spectrometers.

## X-ray spectroscopy

Resonant inelastic X-ray scattering experiments were carried out at endstation ID32-ERIXS, ESRF (Grenoble), with a combined energy resolution of 34 meV at the Ce $M_5$ resonance ($h\nu \approx 882$ eV)[34]. The samples were fractured in situ under ultra-high vacuum conditions. Data was collected in cycles of 5 min for 7–8 h per spectrum and processed using the RIXS Toolbox suite[35]. Data shown in the main text is measured with incident polarization in the scattering plane, without polarization analysis of the scattered beam. The polarization analysis of the scattered beam[36] is demonstrated in Supplementary Note 6[31]. Ce $M$ edge X-ray absorption spectra were obtained during the RIXS experiment (total electron yield mode) and were confirmed by additional measurements in fluorescence yield mode measured at BL29/BOREAS, ALBA (Barcelona), see ref. 31.

## Inelastic neutron scattering

Neutron time-of-flight neutron spectra of $CePd_3$ were obtained on ARCS, SNS (Oak Ridge) at an incident energy of $E_i = 400$ meV. The same 20 g single crystal and experimental procedures were used as in ref. 6.

## Computational models

The temperature evolution of the $^2F_{5/2}$ and $^2F_{7/2}$ excitations (Fig. 4a, black solid lines) was inferred by the momentum-averaged dynamic susceptibility $\chi''(\omega)$ in the Anderson impurity model (AIM) computed with the non-crossing approximation[37], as described in our earlier work[38] and discussed in ref. 31. The NCA/AIM description assumes a single $f$-level (with large orbital degeneracy), a constant hybridization with a Gaussian shape (fixed width) of the conduction band.

The DFT+DMFT electronic structures of $CePd_3$ shown in Fig. 1 were obtained using the Wien2K package[39,40] and the strong-coupling continuous-time quantum Monte Carlo impurity solver[41–44], details can be found in ref. 31. The cerium site energy was shifted by the double-counting term $\mu_{dc}$, which corrects for the $f$–$f$ interaction inherent in the LDA results. The $\mu_{dc}$ value was chosen to reproduce the experimental XAS [cf. Fig. 2d], as well as the electronic spectral function $A(k, \omega)$ observed in direct/inverse photoemission and ARPES data of $CePd_3$, as illustrated in Figs. 1f and 2b and refs. 31, 45, 46. We employed a diagonalization solver with a configuration-interaction scheme for evaluating the RIXS intensities using the Kramers-Heisenberg term from the AIM with the DFT+DMFT hybridization densities (Fig. 2d, e). The AIM-RIXS calculations of the density of states, XAS and RIXS spectra were performed in analogy to our earlier work[47] and are discussed in detail in ref. 31.

## Data availability

The data generated in this study have been deposited in the OpARA database of TU Dresden under accession code 123456789/5706.

## Code availability

All numerical codes in this paper are available upon reasonable request to the authors.

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

## Acknowledgements

We are grateful for provision of experimental time at beamline ID32 of the European Synchrotron Radiation Facility (ESRF), Grenoble, France, at beamline BL5-2 of the Stanford Synchrotron Radiation Lightsource (SSRL) of the SLAC National Accelerator Laboratory, operated by the U.S. Department of Energy (DOE), Office of Science, Office of Basic Energy Sciences (BES) under Contract No. DE-AC02-76SF00515, at beamline 4.0.3. of the Advanced Light Source, a U.S. DOE Office of Science User Facility under contract no. DE-AC02-05CH11231, at beamline UE112_PGM-2b-1^3 of the BESSY-II electron storage ring operated by the Helmholtz-Zentrum Berlin für Materialien und Energie (HZB), at beamline BL29 of the ALBA Synchrotron, Spain, and at the ARCS spectrometer at the Spallation Neutron Source, a DOE Office of Science User Facility operated by the Oak Ridge National Laboratory. Work at Los Alamos National Laboratory was performed under the auspices of the US Department of Energy, Office of Basic Energy Sciences, Division of Materials Sciences and Engineering. M.C.R. is grateful for support through the LANL Director's Fund and the Alexander von Humboldt Foundation. J.-X.Z. was supported by the Center for Integrated Nanotechnologies, a DOE BES user facility. J.K. was supported by the Fonds zur Förderung der wissenschaftlichen Forschung (FWF) through QUAST-FOR5249 (project I 5868-N). A.H. was supported by JSPS KAKENHI Grant Number 21K13884 and 21H01003. K.-H.A. was supported by the Czech Science Foundation Project Grant Number 19-06433S. A.D.C. was partially supported by the U.S. Department of Energy, Office of Science, Basic Energy Sciences, Materials Sciences and Engineering Division. A.A. acknowledges the financial support of the Deutsche Forschungsgemeinschaft DFG under project SE1441/5-2. Work at TU Dresden was supported by the Deutsche Forschungsgemeinschaft through the CRC 1143 and the Würzburg-Dresden Cluster of Excellence ct.qmat (EXC2147, Project ID 390858490). The work by F.H. and M.J. was supported through a Hans Fischer fellowship of the Technische Universität München-Institute for Advanced Study, funded by the German Excellence Initiative and the European Union Seventh Framework Programme under Grant agreement No. 291763. Special thanks are due to Daniel Mazzone, Johan Chang, Pascoal Pagliuso, Jason Hancock, and Peter Riseborough for insightful discussions.

## Author contributions

M.C.R., M.J., F.R., and J.M.L. designed the research. M.C.R., E.B., D.D.B., and K.J.M. synthesized the samples. M.C.R. planned and conducted the RIXS and ARPES measurements, with assistance from M.J., A.A., and F.H.; K.K. and J.D.D. performed additional RIXS, XAS, and ARPES measurements. A.D.C. and J.M.L. planned and performed the neutron measurement. Instrument scientists K.K., J.D.D., D.-H.L., M.H., E.R., and M.V. enabled and assisted in experiments. A.H., J.K., K.-H.A., J.-X.Z., and C.H.B. performed calculations and advised on the theoretical interpretation of the data. M.C.R. analyzed the data. M.C.R. and M.J. wrote the manuscript with feedback from all authors.

## Competing interests

The authors declare no competing interests.
