## [Peer Review File · Nature Communications]

Kondo quasiparticle dynamics observed by resonant inelastic x-ray scatteringReviewer #1 (Remarks to the Author):

This paper studied the evolution of Kondo quasiparticles in the prototype Kondo material, CePd₃, as a function of temperature measured at different energy scales and momentum resolutions by combining the RIXS measurements and the DFT+DMFT calculations. Both experimental and theoretical methods are the state-of-the-art ones to address the fundamental questions of how the strongly correlated nature and the hybridization of Ce 4f electrons with environments changes at different temperatures between the Kondo temperature and the coherence temperature. The scientific results and data are novel and clearly presented. Therefore, I recommend the publication and encourage the authors to consider the following technical comments to be addressed.

1. In the supplemental material, the authors explain that the DFT+DMFT calculation is performed by two steps. First, the DFT+SO calculation with the GGA functional is performed. And then, DMFT with the CTQMC impurity solver is performed by solving the Wannier Hamiltonian projected from the DFT bands. Therefore, it looks like the charge-self-consistent calculation in which the DFT charge is updated from DMFT is not considered. The authors should clarify this and explain why. In principle, it is possible that the energy scales of the SO and CF interactions (E_{SO} , E_{CF}) can be changed when they are computed from the modified DFT charge.

2. Continuing from the first question, it is not clear to me how the E_{SO} and E_{CF} are computed from different CF ground states. As the authors stated in the paper, either the CF ground states or the J states are always somewhat mixed to each other. Moreover, the Wannier functions are usually defined in the $|L,S\rangle$ basis and they need special attentions to be treated as the correlated basis. The authors should explicitly write the Wannier Hamiltonian of 4f electrons and the basis functions they used in the DMFT calculations. Also, they should explain how the DMFT correlated basis function is obtained or transformed from the $|L,S\rangle$ basis defined in the Wannier functions. It would be also helpful to draw the energy level diagram of different orbital ground states.

3. It is very interesting to show that the different double counting corrections can change the f-electron spectra near the Fermi energy dramatically at different temperatures (Fig. S4). The authors should also plot the corresponding DMFT self-energies in both real and imaginary parts to explain what is the origin of this dramatic change due to the double-counting.

4. Because the correlated basis functions are mixed to each other, it is important to check if the off-diagonal terms of the DMFT hybridization function are important or not. It is not explained if these off-diagonal terms are included in CTQMC or AIM-RIXS calculations. They should also mention if the CTQMC calculation is free from the minus-sign problem for the low-temperature calculations.

Reviewer #2 (Remarks to the Author):

The authors (Rahn et al.) combine ARPES, RIXS, XAS, and DFT+DMFT calculations to study the excitations in the Kondo lattice CePd₃ for $T < T_{coh}$ (phase with strongest hybridization between 4f electrons and conduction electrons) and the $T > T_{coh}$ phase (normal Kondo phase). They detected two spin-orbit excitations: $J = 5/2$ and $J = 7/2$ between 0 – 250 meV. The excitations display hardening when decreasing the temperature and dispersion with Q , consistent with DFT+DMFT calculations. From the temperature dependence of these peaks, the authors extracted a microscopic coherent temperature of 150K compatible with other techniques. I think these results are meaningful and important to understand the mechanism of coherent coupling between f-electrons and conduction electrons in Kondo lattice. Moreover, one must praise the multi-technique approach that the authors used to tackle this problem.

I think the work is solid as the authors combined different techniques (ARPES, RIXS, INS, DFT+DMFT) to study this compound. The data are overall consistent, and the presentation is generally clear. They take the advantage of RIXS and performed an orbital-specific analysis of the microscopic spin-orbit excitations of the coherent phase, which cannot be resolved by other techniques. They chose CePd3 due to its Kondo and coherent temperature that can be easily accessed experimentally. My only concern is how these data can be generalized to other Kondo systems, but I do not believe that this concern should preclude publication. The results are important to reveal the Kondo physics of electron systems and deserve the publication in Nature communications after the comments below are addressed.

- In Figure 2(d), it seems the peak position of the panel (d) doesn't match with the panel (e) for $h\nu = E_0$, why is that? For the $J = 7/2$ peak, there is another shoulder at slightly lower energy, what is this excitation?
- In Fig. 2 I am not sure how to interpret the difference between calculations and RIXS data presented in Fig. 2e. While the first peak matches pretty well the second peak is clearly overestimated by the calculations, what causes this discrepancy? Also, why in the plot the authors compare RIXS data at 22K and calculations at 116K. Can they compare with RIXS data at 116K?
- In Fig. 2d the authors claim a broad continuum above 250 meV, is this related to the Pd 4d excitations? It is better to state this more clearly.
- The authors mentioned both quasiparticle excitations are independent of the incident X-ray energy E_0 (page 7). I agree that the peak positions are not changing with the incident energy, but the relative spectral weight clearly changes. Is this an intrinsic property or only a resonant effect?
- Page 7, the authors claim that there is no Q dependence of the 300K RIXS data ($T > T_{coh}$). However, if we closely examine the spectra, we can see both excitation energies and spectral weight changes with Q [Fig. 3a]. Is this only a change of intensity or there is a change as a function of momentum at higher temperature (that is possibly smaller than at lower temperature)?
- It seems that ARPES was only performed at $T = 10$ K, which is below the coherent temperature. If I understand correctly, only this phase will show the quasiparticle bands, which contribute to the Fermi pockets at the Γ and R points. But I think it is important to measure the higher temperature ARPES to prove the Fermi pockets are not there when $T > T_{coh}$, in this case, the observed Fermi pockets are fully contributed to the coherent quasi-particle bands.
- It is better to directly label the T_{coh} in Fig. 4(a) to indicate where the transition is.
- ARPES data lack any k_z resolution, thus one should expect that the k_z dependence is projected over the k_x/k_y plane. Are the authors expecting any broadening if a dispersion is present? Looking at Fig. 1 the electron's dispersion around $\Gamma \rightarrow R$ seems significant. Are the authors neglecting these parts based on the calculations or are there different arguments?
- Concerning the ARPES data in Fig 2c. Can the authors elaborate why the temperature effect leading to the creation of a Kondo peak is intrinsic and not given by thermal broadening?
- Fig. S6 the authors should not use the term direct beam. Unpolarized or not resolved is better.
- Fig. S8, how is the quasi-elastic scattering estimated in the high temperature data. It looks like there is a significant shift from the lower energy peak. This shift might affect the hardening/softening as a function of the temperature. Also being the first peak at 50 meV of higher intensity than the elastic I would like to know more on the process of getting this fitting as peaks are close in energy and a convergent fit seems hard to achieve. What constraints have been used?
- The RIXS polarization selection rule invoked by the authors might not be true. CePd3 is a multiorbital system which imply that angular momentum can be exchanged with both orbitals or spins. Moreover, considerations of the point group symmetry of Ce are required to make these statements. Even if I am convinced that the peak in question is magnetic, I suggest the authors to remove/soften this statement.
- Authors should specify somewhere in text the nominal occupation of Ce. It should be

4f1 if 3+. Probably hybridization leads to additional configurations.

- What is the core origin of the change of hybridization as a function of temperature? Just thermal excitations or also lattice expansion and other structural transitions.
- Page 4 what does bulk measurements mean? Transport? Magnetometry? A more detailed description should be used. This is also repeated in page 5.
- Last sentence of page 4. "This cascade of energy scales makes CePd₃ ideally suited to address both our experimental and computational objectives." It would help readability if the authors could state here the objectives
- In Fig 2a,b there should be a color scale for the ARPES data

Typos

- On page 5, "For $T < T_{coh} \sim 130$ K, coherence sets in, and well-defined quasiparticle bands that cross the Fermi level form pockets at the Gamma and R points (Fig. 1(e))." It seems to me that it should be Fig. 1(f) instead of Fig. 1(e).
- On page 8 (first paragraph), the figure calls of "Fig. 4(b)" should be "Fig. 3(b)".
- there are some inconsistencies in the references that should be fixed

Reviewer #3 (Remarks to the Author):

The manuscript by Rahn et al reports state-of-art RIXS measurements of the archetypal mixed valence compound CePd₃, together with high-resolution ARPES measurements and DFT+DMFT calculations. This work provides a very nice example of how modern RIXS measurements can be used to study the heavy quasiparticle dispersions and dynamics in Kondo lattice systems. While ARPES is traditionally the standard (and direct) method of probing these quasiparticle bands and associated excitations, the current paper shows convincingly that RIXS can be very complementary and useful in such study – it can access unoccupied bands above the Fermi level in a large energy range, which are inaccessible by most other methods, and it is bulk-sensitive compared to ARPES. The current study therefore opens up the exciting opportunity to study quasiparticle dynamics in Kondo lattice systems using RIXS. The interpretation of the RIXS data is well supported by DFT+DMFT calculations, as well as comparison with careful ARPES measurements. A remarkable (new) discovery from the current RIXS study is that the spin-orbit gap of ~ 250 meV remains unchanged after the formation of coherent quasiparticles below T_{coh} , which places strong constraint for future theoretical study. Overall, I found that the paper is very well organized, and the writeup is very clear and easy to follow. The experimental data is of high quality and the interpretation is quite solid. The subject of matter can be of broad interest to the community of strongly correlated systems (particularly for Kondo lattice systems). Therefore, I would recommend publication of this paper in nature communications. However, I do have the following questions that I would recommend the authors to consider before publication of the current paper:

1. Since the current paper involves both ARPES and RIXS measurements (and the focus is both on the quasiparticle bands from Kondo hybridization), I would suggest adding a few sentences in the beginning just to describe briefly how the RIXS spectra could be linked to the quasiparticle bands, i.e., how the dynamic magnetic susceptibility is related to the occupied/unoccupied bands in a coherent Kondo lattice system, for the sake of clarity and general audience. Also it may also be useful to emphasize the difference between INS and RIXS to highlight the importance of the current work.
2. Is the RIXS data in Fig. 2(e) momentum-integrated? The data in Fig. 2(e) is very different from Fig. 2(d), e.g., the $J=5/2$ and $J=7/2$ peak positions. It seems confusing to me – the authors might want to clarify the experimental condition.
3. In page 4 (end of second paragraph) and page 9 (last paragraph): the authors mentioned that RIXS is not limited in energy resolution. This is somewhat confusing: it is well known that the energy resolution of state-of-art RIXS is still not comparable to INS and ARPES at the moment (although it has been improved dramatically in the past 10-20 years). Of course, RIXS is complementary to INS and ARPES and it can offer quasiparticle information in a wide energy range (with different content).
4. The current RIXS results clearly show the momentum dependence of the quasiparticle

excitations at low temperature, when the coherent and dispersive quasiparticle bands develop due to Kondo hybridization. As the authors mentioned in page 4, first paragraph, momentum-dependent/anisotropic hybridization can affect the ground state in a profound manner. In fact, anisotropic hybridization can be most directly observed by ARPES, e.g., in canonical heavy-fermion systems such as Ce-115 (ref 4 and related works) and CeCu₂Si₂. Therefore, I would suggest that the authors add some relevant discussion/literatures on this part, which could help appeal to a broader audience for the paper.

Other minor error:

1. Experimental description, section B photoemission spectroscopy: N edge is 4d-4f transition, not 3d-4f

Nature Communications manuscript NCOMMS-22-00582-T (Rahn *et al.*)

Itemized Response to Reviewers

The corresponding changes are marked in red font in the revised manuscript and SI files.

Reviewer #1 (Remarks to the Author):

This paper studied the evolution of Kondo quasiparticles in the prototype Kondo material, CePd₃, as a function of temperature measured at different energy scales and momentum resolutions by combining the RIXS measurements and the DFT+DMFT calculations. Both experimental and theoretical methods are the state-of-the-art ones to address the fundamental questions of how the strongly correlated nature and the hybridization of Ce 4f electrons with environments changes at different temperatures between the Kondo temperature and the coherence temperature. The scientific results and data are novel and clearly presented. Therefore, I recommend the publication and encourage the authors to consider the following technical comments to be addressed.

We appreciate this favorable assessment of our work, including its methodologies, novelty, and presentation. Reviewer #1 provides a number of qualified comments that show a strong expertise in the computational approaches used in our work. This is very useful input, and we have updated and extended our manuscript and Supplementary Information file accordingly. An in-depth response to each comment is provided below.

1. In the supplemental material, the authors explain that the DFT+DMFT calculation is performed by two steps. First, the DFT+SO calculation with the GGA functional is performed. And then, DMFT with the CTQMC impurity solver is performed by solving the Wannier Hamiltonian projected from the DFT bands. Therefore, it looks like the charge-self-consistent calculation in which the DFT charge is updated from DMFT is not considered. The authors should clarify this and explain why. In principle, it is possible that the energy scales of the SO and CF interactions (E_{SO} , E_{CF}) can be changed when they are computed from the modified DFT charge.

The charge self-consistency (updating the DFT charge) is not implemented in our DFT+DMFT calculation. It does indeed have an effect on the bare SO and CF interactions (the matrix elements of the model Hamiltonian), because the Ce 4f DFT+DMFT and DFT occupancies may somewhat differ. The SO interaction depends primarily on the spherical part of the potential (electrostatic + exchange correlation) on the Ce sites. On the other hand, the CF interaction reflects the charge distribution around Ce sites and the hybridization between Ce 4f and other orbitals (including Pd). The modification of these terms is expected to be minor (since the difference between DFT and DFT+DMFT charge distribution is not large).

The apparent SO and CF interactions (splitting of the bands/levels) in the DFT+DMFT (and DFT) spectra are rather sensitive to the position of bare Ce 4f levels relative to the other bands – this is because, in particular, the CF splitting arises from hybridization rather than from the electrostatic potential. The position of the bare 4f levels depends on the double counting correction, which is not an exact parameter (several different *ad hoc* formulae exist). This uncertainty is not alleviated by enforcing the charge self-consistency, which would add a correction with a far larger uncertainty.

We therefore sacrifice some of the *ab initio* character of our theory and instead adjust one key parameter (the double-counting correction) to achieve consistency with all measured/reported spectroscopies (ARPES, valence+inverse photoemission spectra, M-edge XAS, RIXS). Combining this

adjustable double-counting parameter and enforcing charge self-consistency would be contradictory. We have added a statement to Section 4 of the Supplementary Information file to clarify this point.

2. Continuing from the first question, it is not clear to me how the E_{SO} and E_{CF} are computed from different CF ground states. As the authors stated in the paper, either the CF ground states or the J states are always somewhat mixed to each other. Moreover, the Wannier functions are usually defined in the $|L,S\rangle$ basis and they need special attentions to be treated as the correlated basis. The authors should explicitly write the Wannier Hamiltonian of 4f electrons and the basis functions they used in the DMFT calculations. Also, they should explain how the DMFT correlated basis function is obtained or transformed from the $|L,S\rangle$ basis defined in the Wannier functions. It would be also helpful to draw the energy level diagram of different orbital ground states.

In the revised Supplementary Information file, we show explicitly the local Hamiltonian (input to our DMFT calculations) and plot the Wannier functions constructed from the DFT result. We adopted the $16,7,8$ basis in projecting the Ce 4f bands onto the tight-binding model. These Wannier functions are constructed directly as spinors (with different up and down components) with a given 1-symmetry and thus no transformation from $|L, S\rangle$ is necessary. Transformation to $|L, S\rangle$ basis is of course possible.

3. It is very interesting to show that the different double counting corrections can change the f-electron spectra near the Fermi energy dramatically at different temperatures (Fig. S4). The authors should also plot the corresponding DMFT self-energies in both real and imaginary parts to explain what is the origin of this dramatic change due to the double-counting.

We have added the a plot illustrating the self-energies to Section 4 of the Supplementary Information file (new Figure S6). The strong dependence on the double-counting, which translates in the position of the bare 4f level, reflects physics of a very asymmetric Anderson impurity model (the upper Hubbard band lies at ~ 5 eV, while lower Hubbard band is close to the chemical potential). In the atomic language, the f^0 state is rather close to the f^1 ground state and thus even a small absolute shift of the bare 4f levels results in a large relative change of $E(f^1) - E(f^0)$. This change also affects the Kondo impurity temperature. Therefore the low-energy f-electron spectra with different double-counting values show different temperature evolution.

4. Because the correlated basis functions are mixed to each other, it is important to check if the off-diagonal terms of the DMFT hybridization function are important or not. It is not explained if these off-diagonal terms are included in CTQMC or AIM-RIXS calculations. They should also mention if the CTQMC calculation is free from the minus-sign problem for the low-temperature calculations.

As illustrated in the new Fig. S4 of the Supplementary Information file, the off-diagonal terms of the DMFT hybridization function are small or even zero in the adopted basis. This is because the Wannier functions are the irreducible representations of the Ce site symmetry group (cubic). Consequently, off-diagonal terms are allowed only between states belonging to the same (two 1γ or two 1δ) representations.

In fact, it is intuitive that the off-diagonal hybridization in the allowed pairs is small, given that the two 1γ (or 1δ likewise) states are well split by the spin-orbit coupling (i.e. pairs of $J=7/2$ and $J=5/2$ eigenstates in the limit of no crystal field). It is therefore a safe approximation to neglect these in our CT-QMC and AIM-RIXS calculations. Furthermore, since we deal with only the diagonal hybridization function and density-density term in the Coulomb interaction, our CT-QMC is sign-problem free. We have revised the Supplementary Information file to include this information.

Reviewer #2 (Remarks to the Author):

The authors (Rahn et al.) combine ARPES, RIXS, XAS, and DFT+DMFT calculations to study the excitations in the Kondo lattice CePd₃ for $T < T_{\text{coh}}$ (phase with strongest hybridization between 4f electrons and conduction electrons) and the $T > T_{\text{coh}}$ phase (normal Kondo phase). They detected two spin-orbit excitations: $J = 5/2$ and $J = 7/2$ between 0 – 250 meV. The excitations display hardening when decreasing the temperature and dispersion with Q, consistent with DFT+DMFT calculations. From the temperature dependence of these peaks, the authors extracted a microscopic coherent temperature of 150K compatible with other techniques. I think these results are meaningful and important to understand the mechanism of coherent coupling between f-electrons and conduction electrons in Kondo lattice. Moreover, one must praise the multi technique approach that the authors used to tackle this problem.

I think the work is solid as the authors combined different techniques (ARPES, RIXS, INS, DFT+DMFT) to study this compound. The data are overall consistent, and the presentation is generally clear. They take the advantage of RIXS and performed an orbital-specific analysis of the microscopic spin-orbit excitations of the coherent phase, which cannot be resolved by other techniques. They chose CePd₃ due to its Kondo and coherent temperature that can be easily accessed experimentally. My only concern is how these data can be generalized to other Kondo systems, but I do not believe that this concern should preclude publication. The results are important to reveal the Kondo physics *f* electron systems and deserve the publication in Nature communications after the comments below are addressed.

We are grateful for this positive appraisal of our manuscript. Our study is indeed based on an unusually large number of complex experiments at large-scale facilities. It is very rewarding that its relevance is being recognized and that the manuscript is received as accessible and readable.

The reviewer's concern regarding the generalization to other Kondo systems is well taken. Without doubt, work on other intermediate valence materials will require an individual assessment of the electronic structure (ideally, also in experiment and theory). The RIXS response will depend on many aspects (such as the size and shape of the quasiparticle Fermi surface and the relative size of spin-orbit and crystal field effects). It will be of great interest to observe and interpret these differences. The purpose of our paper is to demonstrate that such material-specific knowledge (beyond impurity models) of the phenomenon is now accessible. We believe that this approach holds an enormous potential for fundamental insights into the local-itinerant dichotomy of electronic many-body systems.

In response to the itemized comments of Reviewer #2, we have revised several aspects of the paper. These changes are discussed below.

- In Figure 2(d), it seems the peak position of the panel (d) doesn't match with the panel (e) for $h\nu = E_0$, why is that? For the $J = 7/2$ peak, there is another shoulder at slightly lower energy, what is this excitation?

Many thanks for pointing out this discrepancy. We had in fact improperly labeled the data in Fig. 2 (d,e). The excitations appear at different energies because the spectra were recorded at different temperatures (Panel (d): 22 K / Panel (e): 300 K). The purpose of Panel (d) is to provide an overview of a raw RIXS spectrum and its different contributions, which are best resolved and easier to illustrate at low temperature. On the other hand, the purpose of Panel (e) is to illustrate which

aspects of the spectra can be captured by the AIM-RIXS computational approach - specifically, its photon energy dependence (see below).

Another difference between the two presentations of the data is that in Panel (e), the elastic peak (whose intensity is not well controlled in RIXS) has been subtracted from the data. This is meant to simplify the comparison to the calculated spectra in Panel (e). For full transparency of this analysis, we provide detail views of the assigned intensities in the Supplementary Information file, see Fig. S11 Panel (7a). This subtraction of the elastic line is very well controlled, because it is an ideal Gaussian with the width of the instrumental resolution.

There may indeed be a very weak shoulder on the low-energy side of the $J=7/2$ peak. As indicated in Fig. S11 Panels (1a)-(6a), we assign this feature to a changeover of intensities between distinct $J=7/2$ - like excitations. Our study shows that this "fine structure" of the spin-orbit transition disperses, i.e. it originates in the emergent band-character of the spin-orbit manifold at low temperatures.

- In Fig. 2 I am not sure how to interpret the difference between calculations and RIXS data presented in Fig. 2e. While the first peak matches pretty well the second peak is clearly overestimated by the calculations, what causes this discrepancy? Also, why in the plot the authors compare RIXS data at 22K and calculations at 116K. Can they compare with RIXS data at 116K?

This is useful feedback - we concede that the key message of Fig. 2(e) was not easily accessible in our original submission.

This panel is meant to illustrate both the success and the limitations of state-of-the-art computational models. For instance, the calculation overestimates the SO gap, likely because the sampling of the conduction bath is too coarse to accurately capture the subtle effect of the f - d hybridization. The position of the $J=7/2$ -like excitation therefore consistently appears at higher energies.

On the other hand, the calculation successfully reproduces the resonant character of the different excitations, which becomes evident in this photon energy dependence (Section 5 of the Supplementary Information file provides more detail): The $J=5/2$ and $J=7/2$ transitions are Raman-like (they appear at a fixed energy transfer). On the other hand, hybridization allows RIXS to engage with the itinerant bath of Pd $4d$ bands. These contribute the fluorescence-like tail whose excitation energy increases with photon energy.

To emphasize these aspects, we have added explanations to the relevant section of the manuscript (p. 7, 3rd paragraph) and to the caption of Fig. 2. We have also added colored markers to Fig. 2(e) to emphasize the fluorescence peaks. Moreover, added a new Figure S8 to Section 5 of the Supplementary Information file, which illustrates the contribution of Pd $4d$ states in our DFT+DMFT calculation (see also answer to next comment).

Data of the photon energy dependence is unfortunately only available for 300 K, but since the algorithm is not well suited to capture the lattice-coherence (low T specific) effects, this is not essential.

- In Fig. 2d the authors claim a broad continuum above 250 meV, is this related to the Pd $4d$ excitations? It is better to state this more clearly.

Indeed, the continuum excitations are due to the hybridization with Pd $4d$ conduction bands. In M-edge RIXS process, a Ce ($3d$) core electron is nominally excited to the $4f$ -manifold. In CePd₃, the

strong hybridization with the metallic environment allows these excitations to “escape” into the broad Pd 4d conduction bands (see p.7, 3rd paragraph).

The characteristic shift of this fluorescence tail to higher energy transfers with higher photon energies is one of the key aspects in that is adequately captured by the AIM-RIXS model. This demonstrates that the itinerant character of this *f*-electron system is adequately represented in our model of the electronic structure (aside from the fact that only a limited number of continuum levels can be included in the calculation).

We have included an explicit reference to Pd 4d bands in the caption to Fig. 2. To illustrate these states we have also added a new plot to Section 5 of the Supplementary Information file (new Fig. S8).

- The authors mentioned both quasiparticle excitations are independent of the incident X-ray energy E_0 (page 7). I agree that the peak positions are not changing with the incident energy, but the relative spectral weight clearly changes. Is this an intrinsic property or only a resonant effect?

The scattering cross section of the RIXS excitation is maximal at the photon energy where core electrons are directly excited into this state. I.e. the transitions into $J=7/2$ states resonate 0.25~0.30 eV above the $J=5/2$ states. In reference to the previous comment, this is the key difference from fluorescence excitations of itinerant states, which *can be enhanced at arbitrary energy transfers*. In that sense, the relative spectral weight is dominated by the “resonant effect” (the denominator of the Kramers-Heisenberg term). On the other hand, the potential intensity of each transition does depend on the intrinsic properties of the sample (as encoded in the transition matrix elements in the numerator of the Kramers-Heisenberg term).

- Page 7, the authors claim that there is no Q dependence of the 300K RIXS data ($T > T_{\text{coh}}$). However, if we closely examine the spectra, we can see both excitation energies and spectral weight changes with Q [Fig. 3a]. Is this only a change of intensity or there is a change as a function of momentum at higher temperature (that is possibly smaller than at lower temperature)?

Indeed, the data does not let us exclude the possibility that small variations in the spectra persist even well above T_{coh} . We would say that at 300 K our data shows no evidence for dispersion, but yes, there is a variation of relative intensities above the statistic error. And in fact, the DMFT calculation at 400 K (Fig. 1e) does suggest that *f* states at this temperature are still far from a single-ion multiplet structure.

Our important finding/claim in this context is that, compared to 300 K, the 22 K spectra vary *dramatically*. We have slightly rephrased this passage (bottom of p.7 / top of p.8) and the caption to Fig. 3 to soften our statement regarding the 300K data.

- It seems that ARPES was only performed at $T = 10$ K, which is below the coherent temperature. If I understand correctly, only this phase will show the quasiparticle bands, which contribute to the Fermi pockets at the Γ and R points. But I think it is important to measure the higher temperature ARPES to prove the Fermi pockets are not there when $T > T_{\text{coh}}$, in this case, the observed Fermi pockets are fully contributed to the coherent quasi-particle bands.

We performed ARPES at temperatures between 1.6 K and 300 K – the temperature dependence of the Fermi surface pockets is summarized in Fig. 2(c). As suggested by the reviewer, the *f* electronic weight at the Fermi surface has diminished almost completely at 300 K. This can be recognized nicely

by comparison a 300 K gold spectrum [gray line in Fig.2(c)]. This trend is in fact in good agreement with the DMFT results shown in Fig 1 (e,f).

- It is better to directly label the Tcoh in Fig. 4(a) to indicate where the transition is.

We have added such a label.

- ARPES data lack any k_z resolution, thus one should expect that the k_z dependence is projected over the k_x/k_y plane. Are the authors expecting any broadening if a dispersion is present? Looking at Fig. 1 the electron's dispersion around $\Gamma \rightarrow R$ seems significant. Are the authors neglecting these parts based on the calculations or are there different arguments?

As the referee correctly states, the bands of CePd_3 do of course disperse in the k_z direction (just as they do in k_x and k_y). Due to the finite lifetime of the core hole, we anticipate significant k_z broadening in this 3d compound. Consequently, the ARPES data presented in Fig. 2(a) in the k_x - k_y plane is averaged over all k_z . In such cases it is known that the intensity will be maximum at extrema in E vs k_z . It is for this reason that the electron pockets around both Γ and R can be observed for a single photon energy even though they are located at different values of k_z . The dispersive features between Γ and R are difficult to resolve because k_z varies along this momentum cut. Instead, features along Γ -X can be identified because all points at $k_z=0$ must be extrema by symmetry.

- Concerning the ARPES data in Fig 2c. Can the authors elaborate why the temperature effect leading to the creation of a Kondo peak is intrinsic and not given by thermal broadening?

For states below the Fermi level, the reduction in photoemission intensity can vary between 0% as $T \rightarrow 0$ and 50% as $T \rightarrow T_{\text{Co}}$. As can be seen in Fig. 2(c), our peak intensity varies from greater than 3 to less than 0.5 from base temperature to 300 K. A factor of >6 change can not be explained by the thermal smearing of the Fermi Dirac distribution, which at most can give a factor of two change.

- Fig. S6 the authors should not use the term direct beam. Unpolarized or not resolved is better.

We have adapted the figure (new Fig. S9) accordingly.

- Fig. S8, how is the quasi-elastic scattering estimated in the high temperature data. It looks like there is a significant shift from the lower energy peak. This shift might affect the hardening/softening as a function of the temperature. Also being the first peak at 50 meV of higher intensity than the elastic I would like to know more on the process of getting this fitting as peaks are close in energy and a convergent fit seems hard to achieve. What constraints have been used?

This is a valid concern. In RIXS, the intensity of the elastic line is not well controlled, as it may depend on the crystal quality (diffuse scattering from defects), the surface quality, and unresolved low-energy phonons. As the excited 4f states are highly localized, it is relatively safe to assume that phonons do not make a major contribution to the RIXS cross section of CePd_3 (they are certainly not observed in RIXS studies of other Ce compounds).

We therefore infer the intensity of the elastic line from a fit to the lowest temperature (20 K) data, where it is well defined. This contribution is then held fixed in the fits of higher temperature data.

Anti-Stokes (photon energy gain) intensity is inferred without additional parameters. This fitting procedure is described in detail Section 10 of the Supplementary Information file.

Crucially, it must be noted that these fits are not used to infer quantitative information like peak positions and intensities. This would not be meaningful, because any realistic model would need to be based on the dispersive f -band structure (currently beyond the state-of-the-art). The purpose of our phenomenological analysis is merely to illustrate the changeover (around T_{coh}) between two lineshapes, which is evident even in the raw data. In the temperature range of this changeover, peak positions were indeed held fixed to allow convergence. We have added an explicit explanation of these circumstances to Section 10 of the Supplementary Information file.

- The RIXS polarization selection rule invoked by the authors might not be true. CePd₃ is a multiorbital system which imply that angular momentum can be exchanged with both orbitals or spins. Moreover, considerations of the point group symmetry of Ce are required to make these statements. Even if I am convinced that the peak in question is magnetic, I suggest the authors to remove/soften this statement.

This comment refers to the bottom paragraph of p.6 and the analysis presented in Section 3 of the Supplementary Information file. The reviewer is correct to point out that there may be both inter and intraband excitations that involve a change of orbital angular momentum $\Delta=1$ but no spin flip. This is a fair point – a blanket statement is indeed not possible. We have followed the advice and have removed the claim in both manuscript and SI file.

- Authors should specify somewhere in text the nominal occupation of Ce. It should be $4f^1$ if $3+$. Probably hybridization leads to additional configurations.

This is correct. Magnetic Ce ($3+$) nominally has an f -occupation number $n_f=1$, but hybridization effectively decreases this value (*intermediate valence*). In CePd₃, one obtains $n_f=0.75$ ($3.25+$) by Ce L -edge XAS [Fanelli et al., J. Phys.: Condens. Matter 26 (2014) 225602]. Values of effective valences typically also depend on the method of characterization. Early measurements of bond lengths (lattice parameters had implied values around $n_f=0.55$ ($3.45+$) [Gardner et al., J. Phys. F: Metal Phys., 2, 133 (1974)]. We have added this information in the relevant passage (p.4, bottom paragraph).

- What is the core origin of the change of hybridization as a function of temperature? Just thermal excitations or also lattice expansion and other structural transitions.

This question is in fact the main interest of our work. Crucially, a decrease in thermal fluctuations allows the Kondo coherent entanglement between the localized f -moments and their metallic environment affect the Kondo hybridization at a neighboring site (as the valence fluctuations also determine bond lengths, structural degrees of freedom play at least an indirect role).

The resulting *lattice*-coherence of the nominally local valence instabilities can be seen as the essence of the local-itinerant dichotomy that lies at the heart of other quantum-critical phenomena (not least, Fe- and Cu-based superconductors). A microscopic model of the underlying mechanism has not been achieved since this issue was formulated by Phil Anderson in the 1970ies. We recommend <https://arxiv.org/pdf/1605.06993.pdf> for a highly readable overview of the historical context of these research interests.

- Page 4 what does bulk measurements mean? Transport? Magnetometry? A more detailed description should be used. This is also repeated in page 5.

This point is well taken. We have modified these sections accordingly.

p. 4: “(...) bulk measurements as a function of external tuning parameters like magnetic field” \hat{e} *bulk measurements like magnetotransport and magnetostriction*

p. 5: “(...) has only been inferred from bulk properties like maxima in resistivity and magnetic susceptibility”

- Last sentence of page 4. “This cascade of energy scales makes CePd₃ ideally suited to address both our experimental and computational objectives.” It would help readability if the authors could state here the objectives

This is a good idea. We have added such a statement, “(...) *to understand how crystal field and spin-orbit interactions become coherently incorporated into the Kondo lattice state.*”

- In Fig 2a,b there should be a color scale for the ARPES data

We have added color scales to these panels. We have also added color scales and improved the rendering of the DFT+DMFT calculations in Fig. 1.

Typos

- On page 5, “For $T < T_{\text{coh}} \sim 130$ K, coherence sets in, and well-defined quasiparticle bands that cross the Fermi level form pockets at the Gamma and R points (Fig. 1(e)).” It seems to me that it should be Fig. 1(f) instead of Fig. 1(e).

- On page 8 (first paragraph), the figure calls of “Fig. 4(b)” should be “Fig. 3(b)”.

Many thanks for spotting these typos, which we have now corrected.

- there are some inconsistencies in the references that should be fixed

Thanks for pointing this out. We have corrected incorrect capitalizations and unified the display of “URL” and “Eprint” properties of the items in our BibTex file.

Reviewer #3 (Remarks to the Author):

The manuscript by Rahn et al reports state-of-art RIXS measurements of the archetypal mixed valence compound CePd₃, together with high-resolution ARPES measurements and DFT+DMFT calculations. This work provides a very nice example of how modern RIXS measurements can be used to study the heavy quasiparticle dispersions and dynamics in Kondo lattice systems. While ARPES is traditionally the standard (and direct) method of probing these quasiparticle bands and associated excitations, the current paper shows convincingly that RIXS can be very complementary and useful in such study – it can access unoccupied bands above the Fermi level in a large energy range, which are inaccessible by most other methods, and it is bulk-sensitive compared to ARPES. The current study therefore opens up the exciting opportunity to study quasiparticle dynamics in Kondo lattice systems using RIXS. The interpretation of the RIXS data is well supported by DFT+DMFT calculations, as well as comparison with careful ARPES measurements. A remarkable (new) discovery from the current RIXS study is that the spin-orbit gap of ~ 250 meV remains unchanged after the formation of coherent quasiparticles below T_{coh} , which places strong constraint for future theoretical study. Overall, I found that the paper is very well organized, and the writeup is very clear and easy to follow. The experimental data is of high quality and the interpretation is quite solid. The subject of matter can be of broad interest to the community of strongly correlated systems (particularly for Kondo lattice systems). Therefore, I would recommend publication of this paper in nature

communications. However, I do have the following questions that I would recommend the authors to consider before publication of the current paper:

We are very happy about this highly positive view of our work. We thank the reviewer for emphasizing the novelty of our methodology, its relevance and potential impact. The feedback on accessibility/readability is very rewarding – it has been a challenge to present a large number of results from diverse methodologies in one streamlined manuscript. We are glad to address the reviewer’s comments and have adapted the paper and Supplementary Information file accordingly (see individual responses below).

1. Since the current paper involves both ARPES and RIXS measurements (and the focus is both on the quasiparticle bands from Kondo hybridization), I would suggest adding a few sentences in the beginning just to describe briefly how the RIXS spectra could be linked to the quasiparticle bands, i.e., how the dynamic magnetic susceptibility is related to the occupied/unoccupied bands in a coherent Kondo lattice system, for the sake of clarity and general audience. Also it may also be useful to emphasize the difference between INS and RIXS to highlight the importance of the current work.

We agree that introducing the difference between the different spectroscopies may be very helpful for non-experts. A short discussion of this context had in fact been part of an early draft of our paper. We have now incorporated this again as a short additional paragraph (p.4, bottom).

2. Is the RIXS data in Fig. 2(e) momentum-integrated? The data in Fig. 2(e) is very different from Fig. 2(d), e.g., the $J=5/2$ and $J=7/2$ peak positions. It seems confusing to me – the authors might want to clarify the experimental condition.

Many thanks for spotting this error. The experimental conditions of the data in Fig. 2(e) were not properly stated and there was in fact a mistake in the labeling of this panel.

The spectrum in Fig. 2 (d) was recorded at 22 K at $Q=(0.4,0,0.4)$, while Panel (e) shows data of $Q=(0,0,0.5)$ at 300 K. We have revised Fig. 2 and its caption accordingly. The main purpose of Panel (e) is to demonstrate that the AIM-RIXS calculation captures Raman-like character of the 4f inter-/inband dynamics, next to the fluorescence-like interactions with Pd 4d states. We have added markers (broad arrows) to emphasize this rather subtle point.

3. In page 4 (end of second paragraph) and page 9 (last paragraph): the authors mentioned that RIXS is not limited in energy resolution. This is somewhat confusing: it is well known that the energy resolution of state-of-art RIXS is still not comparable to INS and ARPES at the moment (although it has been improved dramatically in the past 10-20 years). Of course, RIXS is complementary to INS and ARPES and it can offer quasiparticle information in a wide energy range (with different content).

The reviewer is certainly correct - the resolution of RIXS is vastly inferior to both INS and ARPES. Our statement refers to the *present study*, i.e. we mean to say that the intrinsic widths of the relevant excitations are much broader than our RIXS instrumental resolution (32 meV). E.g. the dramatic changes between 300 K and 22 K could even be evidenced with last-generation instruments.

We have modified the phrasing of these passages (“RIXS is not limited” -> “our insights are not limited”) to clarify this point.

4. The current RIXS results clearly show the momentum dependence of the quasiparticle excitations at low temperature, when the coherent and dispersive quasiparticle bands develop due to Kondo

hybridization. As the authors mentioned in page 4, first paragraph, momentum-dependent/anisotropic hybridization can affect the ground state in a profound manner. In fact, anisotropic hybridization can be most directly observed by ARPES, e.g., in canonical heavy-fermion systems such as Ce-115 (ref 4 and related works) and CeCu₂Si₂. Therefore, I would suggest that the authors add some relevant discussion/literatures on this part, which could help appeal to a broader audience for the paper.

We agree that ARPES arguably provides the most direct (single-particle) observation of the lattice Kondo lattice-coherence and that our RIXS work must be placed in the context of this type of complementary measurements.

We have modified and extended the relevant section (p.4, 2nd paragraph) to draw more attention to recent remarkable experimental achievements in 4f ARPES.

Other minor error:

1. Experimental description, section B photoemission spectroscopy: N edge is 4d-4f transition, not 3d-4f

Many thanks for spotting this typo, which we have now corrected.

Reviewer #1 (Remarks to the Author):

The authors addressed properly all concerns and questions raised in the previous review. Therefore, I recommend the publication as the current form.

Reviewer #2 (Remarks to the Author):

I have carefully read the revised manuscript and the authors' replies. I think they are quite objective in their replies. I think the authors addressed my comments well and I am personally satisfied with their answers. I recommend the manuscript for publication in Nature Communications.

I spotted a typo:

Page 7 bottom. It should be DFT+DMFT bath not DFT+DFMT bath

Reviewer #3 (Remarks to the Author):

My questions are well addressed in the revised manuscript - therefore I recommend the paper to be published in nature communcations, for its novelty and broad impact in the field.

Second resubmission - Responses to reviewers' comments

- Reviewer #1 (Remarks to the Author):

The authors addressed properly all concerns and questions raised in the previous review. Therefore, I recommend the publication as the current form.

We are glad that Reviewer #1 was satisfied by our previous response.

- Reviewer #2 (Remarks to the Author):

I have carefully read the revised manuscript and the authors' replies. I think they are quite objective in their replies. I think the authors addressed my comments well and I am personally satisfied with their answers. I recommend the manuscript for publication in Nature Communications.

I spotted a typo:

Page 7 bottom. It should be DFT+DMFT bath not DFT+DFMT bath

We are glad that Reviewer #2 was satisfied by our previous response. Many thanks for spotting this typo, which we have now corrected.

- Reviewer #3 (Remarks to the Author):

My questions are well addressed in the revised manuscript - therefore I recommend the paper to be published in nature communications, for its novelty and broad impact in the field.

We are glad that Reviewer #3 was satisfied by our previous response.